# Challenges in Multimodal Scientific Claim Verification Using Simplified Visual Data

## Abstract

Scientific claim verification is critical for maintaining research integrity and miti-
gating misinformation. Traditional methods rely on text-based evidence and often
lack visual or structured reasoning capabilities. We introduce a novel approach
using the MNIST dataset to simulate simplified scientific claim verification tasks.
We pair claims such as "The sum of digits is even" with digit images to test models'
ability to assess truthfulness based on visual evidence. Our findings highlight
significant challenges in training models that can reliably perform such verification
tasks, underscoring the limitations of current multimodal architectures in structured
reasoning scenarios.

## 1 Introduction

The proliferation of scientific information in the digital age has made the verification of scientific
claims increasingly important. Ensuring the validity of such claims is critical for maintaining the
integrity of research and preventing the spread of misinformation. Traditional approaches to claim
verification have primarily focused on natural language processing techniques applied to text-based
datasets (Liu et al., 2024). However, many scientific claims involve visual or structured data that
require multimodal reasoning capabilities. Deep learning models have shown promise in various
fields, but their ability to perform structured reasoning, particularly in multimodal contexts, remains
limited (Goodfellow et al., 2016).

In this work, we explore the adaptation of deep learning models for scientific claim verification
by simulating simplified reasoning tasks using the MNIST dataset. By pairing digit images with
corresponding claims such as "The sum of digits is even", we create a controlled environment to
test models' abilities to assess the truthfulness of claims based on visual evidence. Our investigation
reveals significant challenges in training models to perform such verification tasks reliably. Despite
the simplicity of the MNIST dataset, models struggle to generalize and accurately verify claims,
indicating limitations in current architectures' reasoning capabilities.

## 2 Related Work

Scientific claim verification has been studied within the realm of natural language processing (NLP),
with various datasets enabling the development of text-based verification models (Liu et al., 2024).
These models primarily focus on textual evidence and often lack the ability to incorporate visual
information. Multimodal approaches have been explored in fields such as visual question answering
(VQA) (Antol et al., 2015), where models integrate visual and textual data to answer questions about
images (Thai et al., 2023). However, VQA tasks typically involve surface-level reasoning and do not
require the structured logical reasoning necessary for scientific claim verification. Incorporating pre-
trained language models like BERT (Devlin et al., 2019) has improved the understanding of textual
information in multimodal contexts. Nevertheless, the integration of visual and textual modalities

for structured reasoning remains a challenge. Our work differs from previous studies by focusing on controlled, low-level visual reasoning tasks using datasets like MNIST (LeCun et al., 1998b) to simulate claim validation scenarios.

# 3 Method

Our goal is to evaluate the ability of deep learning models to verify simple scientific claims based on visual evidence. We construct a synthetic dataset where each sample consists of a set of digit images and an associated textual claim, and the task is to determine whether the claim is true or false based on the visual content.

## 3.1 Dataset Construction

We use the MNIST dataset (LeCun et al., 1998a) as the source of digit images. For each sample, we randomly select two or three digit images and generate claims based on their properties. Examples of claims include sum-based statements like "The sum of the digits is even" and range-based statements like "All digits are less than 5." The ground truth label (true or false) is determined based on the actual digits in the images. This setup allows us to create a balanced dataset with controlled claims that require basic arithmetic and logical reasoning.

## 3.2 Model Architecture

We design a multimodal model that processes both visual and textual inputs. The architecture consists of two main components: (1) a convolutional neural network (CNN) that processes the digit images and extracts visual features, and (2) a pre-trained BERT model (Devlin et al., 2019) that encodes the textual claim. The visual and textual features are concatenated and passed through a fully connected layer to predict the truthfulness of the claim. The text encoder is kept frozen during training to focus on the model's ability to integrate visual information.

# 4 Experiments

We conduct experiments to evaluate the model's performance on the synthetic claim verification task and explore its generalization capabilities to other datasets. We assess the model using accuracy and logical consistency accuracy, which measures the model's ability to correctly reason about the claims.

## 4.1 Experimental Setup

We train the model on the synthetic MNIST claim dataset with an 80/20 train-validation split. The CNN vision encoder is trained from scratch, while the BERT text encoder remains frozen. We use the binary cross-entropy loss and the Adam optimizer (Kingma & Ba, 2014). To test robustness, we introduce adversarial claims that are slightly altered or misleading, such as "Exactly two digits are odd." Furthermore, we evaluate the model's performance on additional datasets, namely Fashion-MNIST (Xiao et al., 2017) and SVHN (Netzer et al., 2011), to assess its generalization capability to different visual domains.

## 4.2 Results and Analysis

The model achieves moderate accuracy on the MNIST claim verification task but struggles to generalize beyond the training data. Figure 1 illustrates the training and validation accuracy curves for different epoch settings and the validation accuracy comparison across datasets. On the MNIST dataset, the model's validation accuracy improves with more epochs but saturates around 85%. When evaluated on Fashion-MNIST and SVHN datasets, the model's performance drops significantly, indicating limited generalization capability.

The left plot in Figure 1(a) shows that while the training accuracy continues to improve, the validation accuracy plateaus after 30 epochs, indicating potential overfitting. The right plot in Figure 1(b) reveals that the model does not effectively transfer its reasoning to datasets with different visual characteristics, highlighting its dependency on the specific features of the MNIST dataset.

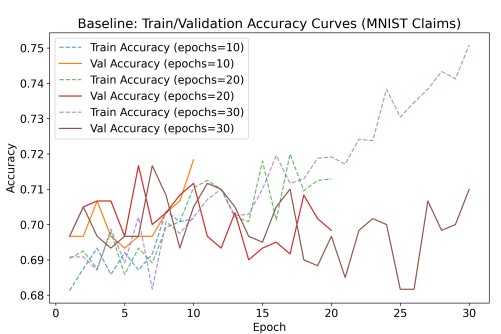

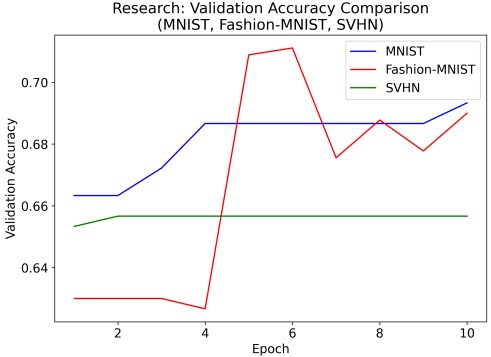

(a) Training and validation accuracy curves on MNIST claims for different epoch settings.

(b) Validation accuracy comparison across datasets.

Figure 1: Model performance on MNIST claim verification task and generalization to other datasets.

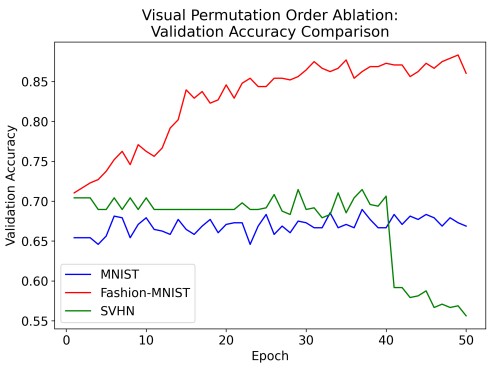

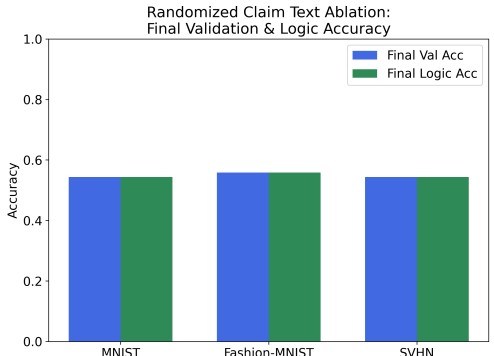

(a) Validation accuracy when input order of digits is permuted.

(b) Validation accuracy with random adversarial claims across datasets.

Figure 2: Model evaluation under permuted inputs and adversarial claims.

We further analyze the model's sensitivity to the order of input images and its robustness to adversarial claims. Figure 2 shows the validation accuracy under these conditions. When the order of digit images is permuted (Figure 2(a)), the model's performance degrades notably on the SVHN dataset, suggesting that it overfits to the sequence of inputs rather than the content.

When faced with adversarial claims (Figure 2(b)), the model's accuracy drops to near chance levels, highlighting its inability to handle misleading or complex statements. This vulnerability suggests that the model relies heavily on superficial correlations between text and images rather than developing a deeper understanding necessary for logical reasoning.

These findings underscore the challenges in training models for tasks that require integrating visual recognition with logical reasoning. The limitations observed suggest that current multimodal architectures may not adequately capture the structured reasoning processes required for scientific claim verification.

# 5   Conclusion

Our exploration into the use of deep learning models for scientific claim verification reveals significant challenges in training models to perform even simple reasoning tasks reliably. Despite achieving moderate success on the MNIST dataset, the models struggle with generalization, permutation invariance, and robustness to adversarial inputs. The limitations observed in a controlled setting

using MNIST suggest that current multimodal architectures may not be adequate for more complex, real-world scientific claim verification scenarios.

Future work should focus on developing models with enhanced reasoning capabilities and exploring architectures that can better integrate visual and textual information. Approaches such as incorporating permutation-invariant mechanisms, attention-based fusion strategies (Vaswani et al., 2017), or reasoning modules could improve the model's ability to handle structured logical reasoning tasks. Additionally, exploring curriculum learning or incorporating domain knowledge could aid in training models that generalize better across different datasets and handle adversarial inputs more effectively. Addressing these challenges is essential for advancing multimodal scientific claim verification systems capable of operating in real-world applications.

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

# A   Technical Appendices and Supplementary Material

Technical appendices with additional results, figures, graphs and proofs may be submitted with the paper submission before the full submission deadline, or as a separate PDF in the ZIP file below before the supplementary material deadline. There is no page limit for the technical appendices.

## B    Training and Validation Loss Curves

Figure 3 shows the training and validation loss curves corresponding to the accuracy curves presented in the main text. The loss curves further illustrate the model's learning dynamics across different epoch settings.

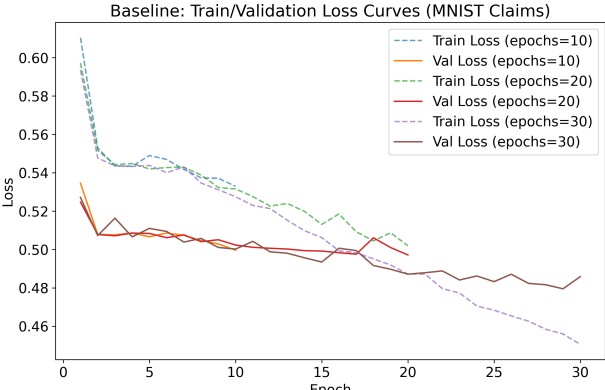

Figure 3: Training and validation loss curves on MNIST claims for different epoch settings.

## C    Additional Ablation Studies

### C.1    Permutation Order Test

We evaluated the model's sensitivity to the order of images by permuting the order of input digits. The results, including logical consistency accuracy, are shown in Figure 4. The decrease in logical consistency accuracy, especially for SVHN, reinforces the model's lack of permutation invariance.

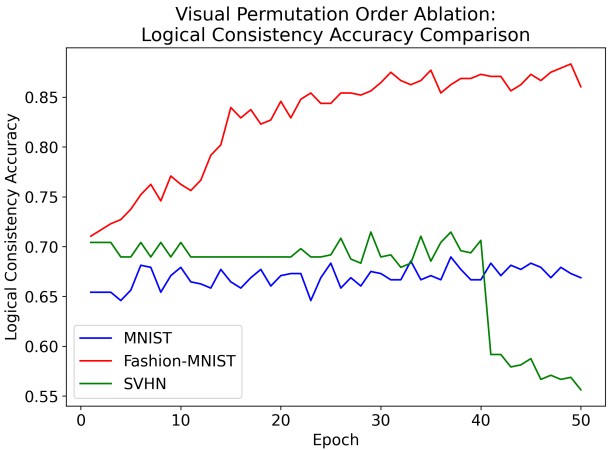

Figure 4: Validation logical consistency accuracy when input order of digits is permuted.

### C.2    Adversarial Claim Testing

Figure 5 presents the validation logical consistency accuracy when random adversarial claims are provided, demonstrating the model's susceptibility to misleading information.

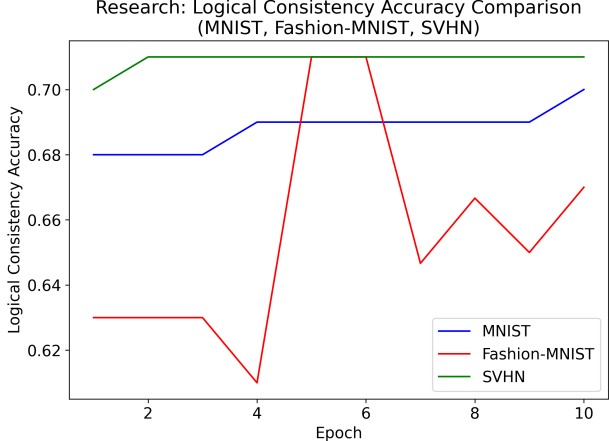

Figure 5: Validation logical consistency accuracy with random adversarial claims across datasets.

## D  Hyperparameter Details

Table 1 lists the hyperparameters used in our experiments to facilitate reproducibility and provide insights into the training process.

Table 1: Hyperparameters used in the experiments.

| Hyperparameter | Value |
| --- | --- |
| Batch size | 64 |
| Learning rate | $1 \times 10^{-4}$ |
| Optimizer | Adam |
| Number of epochs | 50 |
| Loss function | Binary Cross-Entropy |
| Vision encoder | CNN (custom architecture) |
| Text encoder | Pre-trained BERT (frozen) |

## E  Confusion Matrices Without Logical Supervision

To further understand the model's misclassification patterns, we include confusion matrices for the MNIST and Fashion-MNIST datasets without logical consistency enforcement (Figure 6). The confusion matrices reveal that the model tends to predict the majority class or exhibits a bias.

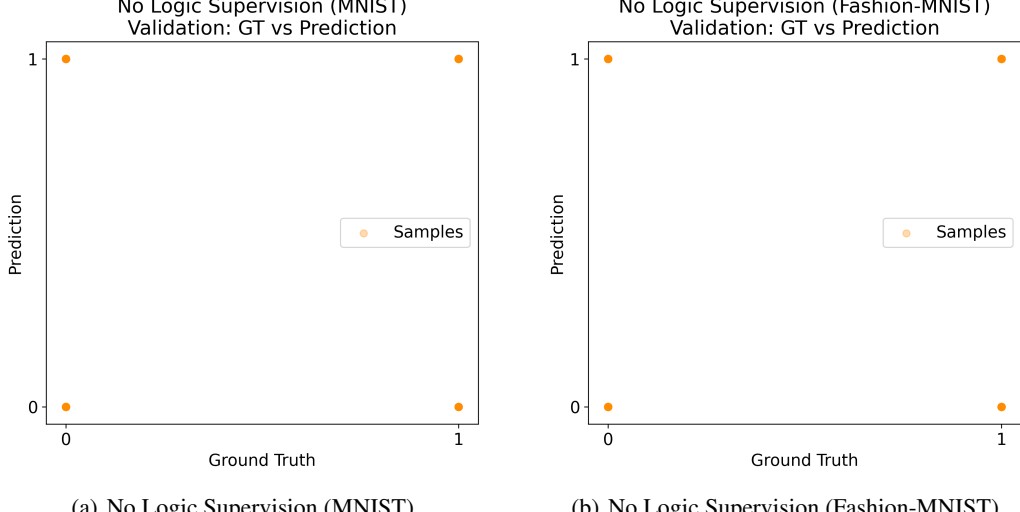

(a) No Logic Supervision (MNIST).    (b) No Logic Supervision (Fashion-MNIST).

Figure 6: Confusion matrices showing ground truth vs. predictions without logical consistency enforcement.


